# Effects of Exercise on Inflammatory Markers in Individuals with Chronic Kidney Disease: A Systematic Review and Meta-Analysis

**DOI:** 10.3390/metabo13070795

**Published:** 2023-06-27

**Authors:** Victor M. Baião, Vinícius A. Cunha, Marvery P. Duarte, Francini P. Andrade, Aparecido P. Ferreira, Otávio T. Nóbrega, João L. Viana, Heitor S. Ribeiro

**Affiliations:** 1Faculty of Health Sciences, University of Brasilia, Brasilia 70910-900, Brazil; victor.baiao@aluno.unb.br (V.M.B.); albuquerquev40@gmail.com (V.A.C.); marveryp@gmail.com (M.P.D.); otavionobrega@unb.br (O.T.N.); heitor.ribeiro@icesp.edu.br (H.S.R.); 2Graduate Program in Pneumological Sciences, School of Medicine, Universidade Federal do Rio Grande do Sul, Porto Alegre 91501-970, Brazil; fandrade@umaia.pt; 3Research Center in Sports Sciences, Health Sciences and Human Development, CIDESD, University of Maia, 4475-690 Maia, Portugal; 4Interdisciplinary Research Department, University Center ICESP, Brasília 71961-540, Brazil; aparecido.ferreira@icesp.edu.br; 5Post-Graduation Program, Santa Úrsula University, Rio de Janeiro 22231-040, Brazil

**Keywords:** kidney failure, dialysis, cytokines, inflammation, resistance exercise, aerobic exercise

## Abstract

Individuals with chronic kidney disease (CKD) have a systemic inflammatory state. We assessed the effects of exercise on inflammatory markers in individuals with CKD. An electronic search was conducted, including MEDLINE. Experimental clinical trials that investigated the effects of exercise on inflammatory markers in individuals with CKD at all stages were included. Meta-analyses were conducted using the random-effects model and standard mean difference (SMD). Subgroup analyses were performed for resistance, aerobic, and combined exercise interventions. Twenty-nine studies were included in the meta-analyses. Exercise interventions showed significant reductions in C-reactive protein (CRP) (SMD: −0.23; 95% CI: −0.39 to −0.06), interleukin (IL)-6 (SMD: −0.35; 95% CI: −0.57, −0.14), and tumor necrosis factor-alpha (TNF-α) (SMD: −0.63, 95% CI: −1.01, −0.25) when compared with the controls. IL-10 levels significantly increased (SMD: 0.66, 95% CI: 0.09, 1.23) with exercise interventions. Resistance interventions significantly decreased CRP (SMD: −0.39, 95% CI: −0.69, −0.09) and TNF-α (SMD: −0.72, 95% CI: −1.20, −0.23) levels, while increasing IL-10 levels (SMD: 0.57, 95% CI: 0.04, 1.09). Aerobic interventions only significantly reduced IL-6 levels (SMD: −0.26, 95% CI: −0.51, −0.01). No significant changes in any inflammatory markers were observed with combined exercise interventions. Exercise interventions are effective as an anti-inflammatory therapy in individuals with CKD compared to usual care control groups. Resistance interventions seem to promote greater anti-inflammatory effects.

## 1. Introduction

Individuals with CKD have a systemic inflammatory condition characterized by persistent alterations in circulating inflammatory markers, which may result in several adverse outcomes, such as cardiovascular disease, protein-energy wasting, anemia, atherosclerosis, bone diseases, morbidity, and mortality [1,2,3,4,5,6]. The persistent chronic systemic inflammation in individuals with CKD may be caused by several factors, including the high production of pro-inflammatory cytokines, oxidative stress, metabolic acidosis, chronic and recurrent infections, and a disorder of adipose tissue metabolism [1,3,7,8].

Inflammatory markers, such as the interleukin (IL) family (e.g., IL-1β, IL-1 receptor antagonist, IL-6), tumoral necrosis factor-alpha (TNF)-α, and c-reactive protein (CRP), are inversely associated with kidney function and positively with albuminuria [1]. Moreover, skeletal muscle is an endocrine organ producing and releasing both pro- and anti-inflammatory cytokines (i.e., myokines). The skeletal muscle contraction releases myokines, which exert specific systemic endocrine effects, modulating the global inflammatory condition [9,10].

Several strategies have been proposed to attenuate the inflammation in CKD, including optimal dialysis treatment and lifestyle modifications [11,12]. Engagement in exercise programs is an example of a lifestyle modification and the effects have been reported to provide improvements or maintenance in inflammatory markers [13,14,15,16]. After acute exercise bouts, there is an increase in circulating levels of IL-6 myokine, which promotes a pro-inflammatory condition; however, the increased myokine IL-6 has a positive and transient effect triggering the release of IL-10, which is an anti-inflammatory cytokine [10,17].

There is scientific evidence showing that different types of exercise interventions might play an important role in attenuating the chronic inflammatory condition in CKD [10,18,19,20,21]. However, no previous systematic review and meta-analysis has synthesized the effects of different exercise interventions on inflammatory markers (i.e., CRP, IL-6, IL-10, and TNF-α) across the wide spectrum of CKD. Thus, the present study aimed to assess the effects of exercise interventions on inflammatory markers in the whole spectrum of CKD.

## 2. Materials and Methods

### 2.1. Registration and Protocol

This study was registered at the International Prospective Register of Systematic Reviews (CRD42020207830). The systematic review was performed using the **PICOS** framework: individuals with CKD (**P**opulation); exercise (**I**ntervention); usual care control (**C**omparison); inflammatory markers (i.e., CRP, IL-6, IL-10, and TNF-α) (**O**utcomes); experimental clinical trials (**S**tudy design). Also, we followed the Preferred Reporting Items for Systematic Reviews and Meta-Analyses (PRISMA) [22] and Cochrane Collaboration recommendation statements [23].

### 2.2. Eligibility Criteria

Only experimental clinical trials, randomized or not, were included. We investigated the effects of different types of exercise interventions that lasted ≥4 weeks and that evaluated the effects of exercise on inflammatory cytokines in CKD adults (≥18 years). Exercise was considered an intervention usually prescribed to improve or maintain fitness and/or health. We included aerobic, resistance, and combined (aerobic and resistance exercise) interventions. Inflammatory cytokines were considered a type of protein that is made by certain immune and non-immune cells and influences the immune system. Due to the enormous number of inflammatory cytokines, we have chosen the most commonly used [1,24]: CRP, high sensitivity (hs)-CRP, IL-6, IL-10, and TNF-α. The following exclusion criteria were considered: animal studies, dietary and/or pharmacological interventions, conference abstracts, thesis, letters to the editor, and case reports.

### 2.3. Search Strategy

A systematic search was performed for two independent authors (V.A.C. and V.M.B) at MEDLINE, Cochrane Central Register of Controlled Trials, and LILACS from inception until January 2021. A search strategy was developed for each database using a combination of free text and controlled vocabulary terms (Appendix A). We used search terms related to CKD, exercise, and inflammatory cytokines. No language and date restrictions were set. The reference list of the final selected articles was consulted to find possible additional studies. An update search was conducted in January 2022.

### 2.4. Study Selection

Two independent authors (V.A.C. and V.M.B) screened titles and abstracts to identify potential studies and judged those to be included after a full-text reading. Disagreements and conflicts were resolved by consensus, and if necessary, a third reviewer was consulted (H.S.R). The duplicate items identified after searching the databases were removed. We performed all selection steps using the Start software (v. 3.4. Beta 03, UFSCar, São Carlos, Brazil).

### 2.5. Data Extraction

The main reviewer (V.A.C) performed data extraction of the selected studies and a double-check was conducted by a second reviewer (V.M.B). The following information was extracted: country, age, sample size, CKD stage, kidney replacement therapy modalities (if applicable), setting, types of exercise, intervention duration and frequency, and inflammatory markers evaluated. The continuous outcome data (i.e., inflammatory cytokines) were extracted to perform the quantitative synthesis. All data were entered into a database on an Excel spreadsheet.

### 2.6. Methodological Assessment

#### 2.6.1. PEDro

The methodological quality of the included studies was scored using the PEDro scale to assess the risk of bias by two independent authors (V.A.C. and V.M.B). PEDro rates clinical trials from 0 (low quality) to 10 (high quality). A score ≥6 was classified as high-quality, while trials with a score <6 were classified as low-quality [25].

#### 2.6.2. Quality of Evidence

The quality of evidence was evaluated according to the Grading of Recommendations, Assessment, Development, and Evaluation (GRADE) criteria [26] and the Cochrane Handbook for Systematic Reviews of Interventions [23]. For each outcome, the quality of evidence was based on five factors: (1) risk of bias; (2) consistency; (3) directness; (4) precision; and (5) publication bias. The levels of evidence were characterized as high, moderate, low, or very low.

### 2.7. Data Analysis

We conducted meta-analyses to determine pooled-effect changes in inflammatory markers pre and post intervention by calculating the standardized mean difference (SMD) between the exercise group and usual care control groups with a 95% of confidence interval (CI). The random effects were considered for the meta-analyses due to the variations in the CKD stages, types of exercises, intervention parameters, and settings. The random effects may incorporate better each study variability and minimize the likelihood of type I errors [27]. Subgroup meta-analyses were conducted for resistance, aerobic, and combined exercise interventions. The heterogeneity was assessed by the I^2^ statistic [23]. An I^2^ statistic of 0% to 40% might not be important, 30% to 60% may represent moderate heterogeneity, 50% to 90% may represent substantial heterogeneity, and 75% to 100% considerable heterogeneity [23]. In addition, meta-regression subgroup analyses were performed for identifying the source of heterogeneity. When there were enough studies to pool, dialysis was compared to non-dialysis individuals, intervention length <16 (median value) versus ≥16 weeks, and sample size ≥41 (median value) versus <41 individuals. Funnel plots were built for visual inspection to assess possible publication bias. All meta-analyses were performed using Review Manager (version 5.4, the Cochrane Collaboration, 2020), and additional statistical analyses were performed with Statistical Package for the Social Sciences (version 26.0, IBM Corp., Armonk, NY, USA).

## 3. Results

### 3.1. Study Selection

The search strategy retrieved 1728 potentially eligible studies (Figure 1). After eligibility criteria, 30 studies [4,13,14,18,19,20,21,28,29,30,31,32,33,34,35,36,37,38,39,40,41,42,43,44,45,46,47,48,49,50] were included and one study did not show enough data to be included in the meta-analysis [44].

### 3.2. Characteristics of the Included Studies

#### 3.2.1. Participants

The studies included in the review were mainly from America (56.7%), followed by Asia (23.3%). All studies represented a total of 1471 individuals with CKD. Hemodialysis was the most prevalent dialysis modality (22 studies: 73.3%). The number of individuals with CKD ranged from 11 to 170, the mean age ranged from 39 to 67 years, and the length of follow-up ranged from 8 to 96 weeks (Appendix A).

#### 3.2.2. Intervention

Resistance (11 studies, 36.7%) and aerobic interventions (12 studies, 40.0%) were the most performed, followed by combined (seven studies, 23.3%). Information about the intensity, frequency, and setting of the exercise interventions may be seen in Appendix A. The exercise intensities were monitored in different ways. Aerobic and resistance interventions were mainly monitored by the Borg scale and the percentage of maximum repetitions, respectively. Regarding the exercise frequency, 90.0% performed ≤3 days per week (25 studies). The most prevalent setting was intradialytic (*n* = 19, 63.3%).

#### 3.2.3. Methodological Quality

Based on PEDro scale, six studies were classified as having a high-quality methodology. Furthermore, 14 studies had a sample loss lower than 85% (Appendix A).

#### 3.2.4. Quality of Evidence

According to the GRADE criteria, the quality of evidence for the outcomes ranged from very low to moderate and can be seen in Appendix A.

#### 3.2.5. Publication Bias

The funnel plots for each outcome did not display overt asymmetries by visual inspection, except for the IL-10 outcome (Appendix A).

#### 3.2.6. Inflammatory Markers

CRP and hs-CRP were evaluated in 70.0% (932 individuals), IL-6 in 56.7% (715 individuals), IL-10 in 39.0% (629 individuals), and TNF-α in 33.3% of the studies (558 individuals) (Appendix A).

### 3.3. Interventions Effects

Table 1 shows the meta-analyses performed through the random-effects model investigating the effects of exercise interventions on CRP, hs-CRP, IL-6, IL-10, and TNF-α levels in individuals with CKD. In addition, Appendix A presents the meta-analyses performed with the fixed-effects model.

#### 3.3.1. Exercise Effects on C-Reactive Protein

CRP and hs-CRP were merged for analysis. Figure 2 shows a significant CRP reduction after exercise interventions (21 studies; SMD: −0.26; 95% CI: −0.45 to −0.08; I^2^ = 44%; very low evidence quality). A significant CRP reduction was also seen after resistance interventions (seven studies; SMD: −0.39; 95% CI: −0.69 to −0.09; low evidence quality). There was no significant change after aerobic and combined exercise interventions (Figure 2).

Subgroup analyses indicated a significant reduction in CRP levels among hemodialysis patients compared to control groups (15 studies; SMD: −0.30; 95% CI: −0.54 to −0.05; *p* = 0.55), but the same was not found in non-dialysis individuals (six studies; SMD: −0.18; 95% CI: −0.45 to 0.08; *p* = 0.55). Exercise interventions that lasted <16 weeks showed a significant reduction (eight studies; SMD: −0.55; 95% CI: −0.94 to −0.17; *p* = 0.03) compared to control groups, whereas exercise interventions ≥16 weeks did not show a significant change (13 studies; SMD: −0.09; 95% CI: −0.27 to 0.09; *p* = 0.03). Interventions with a larger sample size (≥41 versus <41 individuals) did not show a significant effect in CRP (21 studies; SMD: −0.14; 95% CI: −0.35 to 0.07; *p* = 0.91).

#### 3.3.2. Exercise Effects on Interleukin 6

The meta-analysis in Figure 3 showed a significant IL-6 reduction in pooled exercise interventions (16 studies; SMD: −0.29; 95% CI: −0.50 to −0.07; I^2^ = 44% low evidence quality). However, in the subgroup analyses, there were no significant changes after aerobic, resistance, and combined exercise interventions. In subgroup analyses, non-dialysis individuals who performed exercise showed a significant reduction (five studies; SMD: −0.64; 95% CI: −1.01 to −0.27; *p* = 0.02) compared to control groups, but the same was not found in those dialysis individuals (10 studies; SMD: 0.01; 95% CI: −0.39 to 0.41; *p* = 0.02).

Exercise interventions ≥16 weeks showed a significant IL-6 reduction (ten studies; SMD: −0.33; 95% CI: −0.58 to −0.07; *p* = 0.81) compared to control groups, whereas exercise interventions <16 weeks did not show a significant change (six studies; SMD: −0.28; 95% CI: −0.62 to 0.07; *p* = 0.81). Studies with a sample size of <41 individuals showed a significant IL-6 reduction after the exercise interventions (ten studies; SMD: −0.31; 95% CI: −0.55 to −0.06; *p* = 0.95) compared to control groups, while those with ≥41 individuals did not show a significant change (six studies; SMD: −0.29; 95% CI: −0.65 to 0.06; *p* = 0.95).

#### 3.3.3. Exercise Effects on Interleukin 10

The meta-analysis in Figure 4 showed a significant increase in IL-10 after exercise interventions (11 studies; SMD: 0.52, 95% CI: 0.04 to 1.00; I^2^ = 87%; very low evidence quality). In the subgroup analyses, there was a significant increase after resistance interventions (SMD: 0.69, 95% CI: 0.00 to 1.37; very low evidence quality). However, aerobic interventions did not show a significant change (SMD: 0.28, 95% CI: −0.31 to 0.87; very low evidence quality). There was not enough evidence for combined interventions (Figure 4).

There were no significant differences in subgroup analyses according to dialysis and non-dialysis treatments (seven studies; SMD: 0.36; 95% CI: −0.12 to 0.84; *p* = 0.42); ≥16 and <16 weeks of exercise interventions (five studies; SMD: 0.46; 95% CI: −0.14 to 1.05; *p* = 0.82); and sample size of ≥41 and <41 individuals (four studies; SMD: 0.53; 95% CI: −0.12 to 1.18; *p* = 0.95).

#### 3.3.4. Exercise Effects on Tumor Necrosis Factor-Alpha

Figure 5 showed a significant reduction after pooled exercise interventions (11 studies; SMD: −0.48, 95% CI: −0.84 to −0.12; I^2^ = 74%; moderate evidence quality). A significant reduction in TNF-α after resistance interventions was also seen (SMD: −0.72, 95% CI: −1.20 to −0.23; moderate evidence quality), whereas aerobic interventions (SMD: −0.34, 95% CI: −0.84 to 0.16; very low evidence quality) and combined interventions (SMD: −0.02, 95% CI: −1.21 to 1.17; very low evidence quality) did not show significant changes.

In subgroup analyses, both dialysis and non-dialysis subgroups did not show a significant effect in TNF-α after exercise interventions ≥16 weeks (six studies; SMD: −0.47; 95% CI: −1.01 to 0.06; *p* = 0.99). However, exercise interventions <16 weeks (four studies; SMD: −0.47; 95% CI: −0.91 to −0.03; *p* = 0.03) showed a significant TNF-α reduction. In addition, studies with a sample size of ≥41 individuals showed a significant TNF-α reduction (six studies; SMD: −0.63; 95% CI: −1.07 to −0.20; *p* = 0.23) after the exercise interventions, while those with <41 individuals (four studies; SMD: −0.19; 95% CI: −0.76 to 0.38; *p*= 0.23) did not show a significant change between exercise interventions and control groups.

## 4. Discussion

### 4.1. Main Findings

The present systematic review and meta-analyses revealed that exercise interventions have important anti-inflammatory effects in the wide spectrum of CKD. In addition, resistance interventions resulted in greater anti-inflammatory effects compared to other exercise modalities such as aerobic and combined interventions. Subgroup analyses according to the CKD stage and length of intervention revealed that CRP only decreased in dialysis individuals, IL-6 levels only decreased in studies with a larger sample size (≥41 individuals), and TNF-a levels only decreased in the exercise interventions with longer duration (≥16 weeks).

### 4.2. Interventions Effects

#### 4.2.1. C-Reactive Protein

Our meta-analysis revealed a reduction in CRP after pooled exercise interventions, which is important because higher CRP values are associated with mortality risk in individuals with CKD [2,51]. However, the subgroup analysis revealed that only resistance exercise interventions were able to reduce CRP. According to the literature, resistance exercise interventions are more effective than aerobic exercise interventions to promote an increase in strength and muscle mass [52,53]. In addition, there is scientific evidence revealing a negative association of CRP with strength and muscle mass [54].

Also, the subgroup analysis showed a reduction in CRP levels only among dialysis. End-stage kidney disease leads to higher CRP levels due to chronic inflammation [55,56], suggesting that dialysis individuals may exhibit greater sensitivity to changes in CRP levels through exercise interventions. However, non-dialysis patients with lower CRP levels may cause less sensitivity to CRP variations. Corroborating with our findings, the systematic review of Wu et al. [57] revealed that exercise in non-dialysis individuals also did not reduce CRP values.

#### 4.2.2. Interleukin 6

Our meta-analysis showed a significant reduction in IL-6 after pooled exercise interventions. IL-6 is considered an acute anti-inflammatory cytokine, whereas chronically, it may be pro-inflammatory [58]. Muscle mechanical stress generated by acute bouts of exercise increases IL-6 levels [10,17]; however, this is followed by an increase in anti-inflammatory cytokines such as IL-1ra and IL-10 [17,59]. Previous evidence has shown that high IL-6 values are associated with the progression of coronary artery calcification and mortality in dialysis individuals [60]. Thus, we believe that the decrease in IL-6 values promoted by exercise interventions may lead to a chronic improvement in the inflammatory condition in individuals with CKD [10,61], as well as a protective effect on cardiovascular outcomes, such as coronary artery calcification.

Our subgroup analysis showed a significant IL-6 reduction only in non-dialysis individuals. Many factors may act to reinforce the inflammatory state in dialysis individuals (i.e., uremia and dialysis adequacy per se) [11,62]. In addition, according to the study by Dungey et al. [63], the uremia present in dialysis individuals inhibits the pathway for exercise-induced cytokines secretion, i.e., IL-6 myokine. Another important finding of the subgroup meta-regression was that only studies with longer periods of intervention reduced IL-6. So, we believe that individuals with CKD need more time to become adapted to the exercise interventions and experience their benefits.

In addition, our subgroup analysis also revealed that studies with <41 individuals reduced IL-6 more than studies with ≥41 individuals, which may be explained by the fact that 70.0% of studies with ≥41 individuals were performed with dialysis individuals, revealing a possible interference of dialysis in the effect of exercise on inflammatory markers.

#### 4.2.3. Interleukin 10

The IL-10 values increased after pooled exercise interventions, which is a positive effect, as IL-10 is an anti-inflammatory marker that inhibits the production of pro-inflammatory cytokines (e.g., IL-1α, IL-1β, and TNF-α) [17,21,64]. In addition, according to the subgroup analysis, it seems that only resistance interventions were able to increase IL-10 values. The resistance interventions promote a higher hypertrophic effect than aerobic interventions [52], which may induce a higher increase in IL-10 levels mediated by IL-6 myokine production [17,65].

#### 4.2.4. Tumor Necrosis Factor-Alpha

Our results reveled a significant reduction in TNF-α after pooled exercise interventions. This finding may benefit non-dialysis individuals because high levels of TNF-α are inversely associated with kidney function [55]. Furthermore, the interventions subgroup analysis showed a reduction in TNF-α only in resistance interventions. The hypertrophic effect promoted by the resistance exercise interventions caused not only an increase in IL-10 as previously explained but also in the soluble TNF receptor, which inhibits the production of TNF-α [10,17,65]. Another important finding of our subgroup analysis was the fact that, regardless of the duration of the intervention, there was a decrease in TNF-α, revealing that even short periods of intervention may be effective on TNF-α modulation. Possibly, this change has a greater magnitude in the first weeks when the body is adapting to the new stimuli promoted by exercise and, after this initial period, there is an adaptation to the intervention and it therefore no longer brings significant variations, since according to the principle of progressive overload, the body needs training with loads greater than those to which it is adapted [66]. In addition, only studies with ≥41 individuals showed a significant reduction in TNF.

### 4.3. Clinical Applicability

The findings from our systematic review evidence the important role of exercise interventions on inflammatory markers. As we have recently published [67], exercise is becoming part of the routine care of this population and our results here put light on very relevant clinical outcomes. We, therefore, highlight the importance of applying exercise training principles, especially the progression over time [67], to possibly optimize anti-inflammatory effects. Lastly, we believe that more studies are needed to explore and understand the effects of different exercise intensities and volumes on inflammatory markers in individuals with CKD, especially those non-dialysis-dependent and -transplanted.

### 4.4. Strengths and Limitations

To our knowledge, this is the first systematic review and meta-analysis that provides the effects of different types of exercise interventions on many inflammatory markers (CRP, IL-6, IL-10, and TNF-α) in different CKD stages. Our review had a large number of individuals (*n* = 1603) and described the details of each study included, followed a strict methodological standard, and showed the methodological quality of studies and the quality of evidence. Moreover, we explored the causes of heterogeneity by conducting multiple subgroup analyses.

Yet, some limitations in our study must be recognized, such as the high heterogeneity that may have been caused by the wide variation in exercise interventions, different settings and stages of CKD, and the variation among the studies in the methods to measure the inflammatory markers.

## 5. Conclusions

In conclusion, our findings reveal the potential anti-inflammatory effects of exercise interventions in individuals with CKD. Furthermore, resistance exercise may be more effective in reducing TNF-α and CRP levels and increasing IL-10 levels. Thus, resistance exercise interventions must be recognized as the primary exercise type to be prescribed when targeting anti-inflammatory effects in individuals with CKD.

## Figures and Tables

**Figure 1 metabolites-13-00795-f001:**
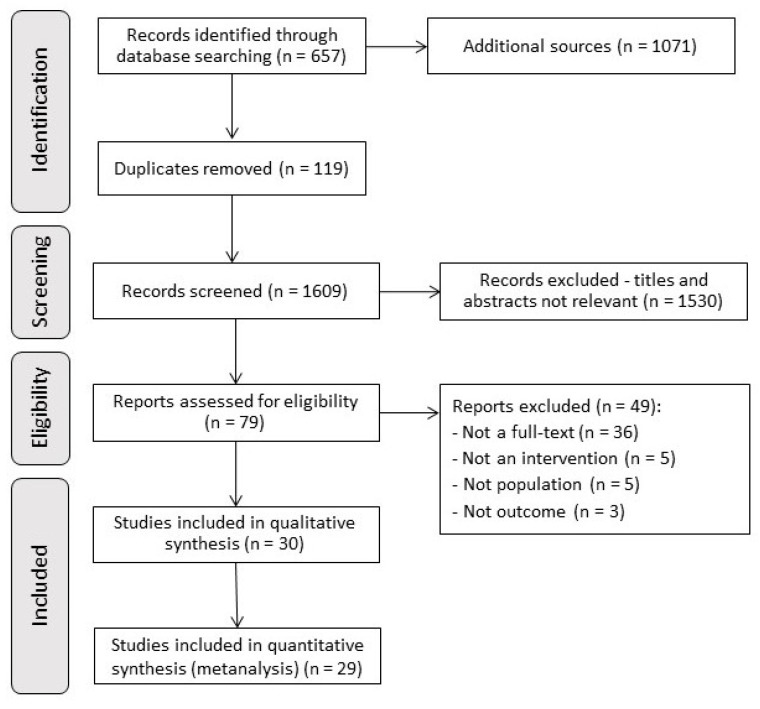
PRISMA flowchart.

**Figure 2 metabolites-13-00795-f002:**
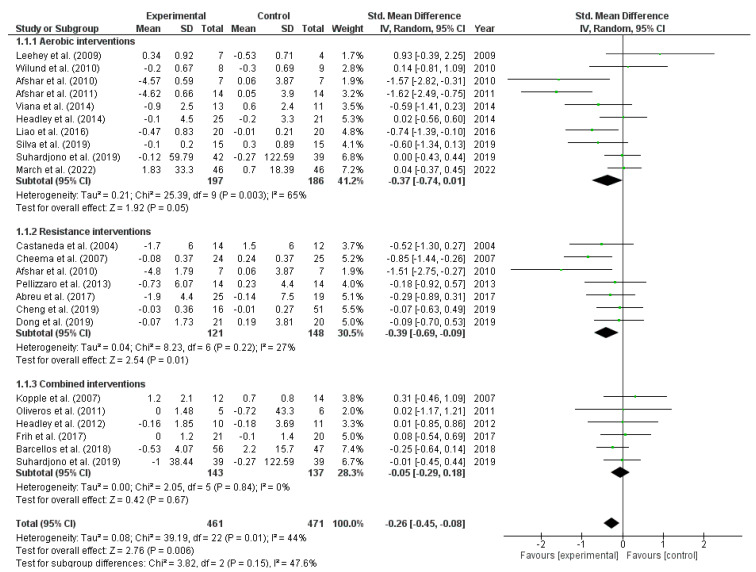
Forest plot of the difference in C-reactive protein (CRP) between exercise interventions and controls.

**Figure 3 metabolites-13-00795-f003:**
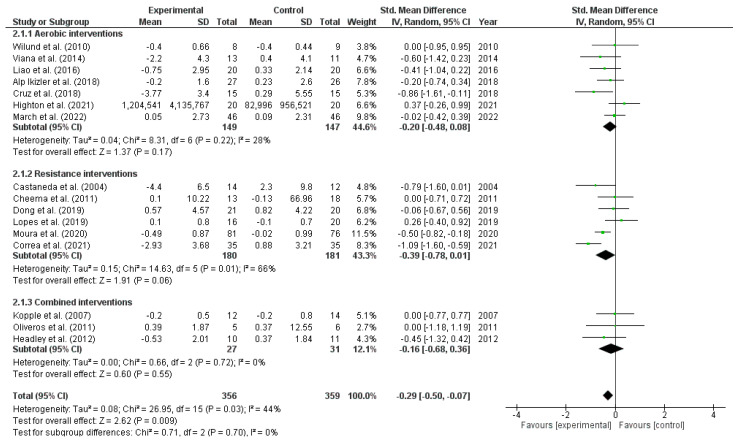
Forest plot of the difference in interleukin-6 (IL-6) between exercise interventions and controls.

**Figure 4 metabolites-13-00795-f004:**
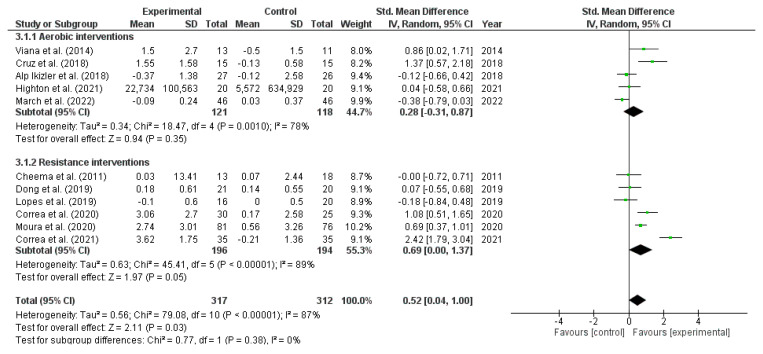
Forest plot of the difference in interleukin-10 (IL-10) between exercise interventions and controls.

**Figure 5 metabolites-13-00795-f005:**
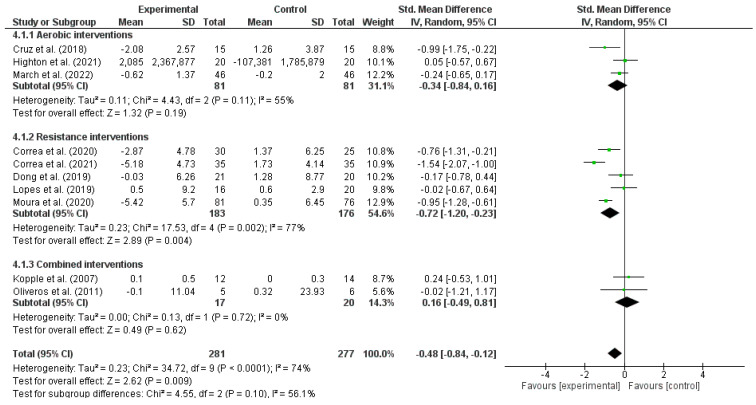
Forest plot of the difference in tumoral necrosis factor-alpha (TNF-α) between exercise interventions and controls.

**Table 1 metabolites-13-00795-t001:** Meta-analyses performed in the review on the effects of exercise on CRP, IL-6, IL-10, and TNF-α levels.

Exercise Intervention	Studies	Individuals	Std. Mean Difference (95% CI)	Heterogeneity (i2, %)
**Pooled exercise interventions**				
CRP	21	932	−0.26 (−0.45 to −0.08)	44
IL-6	16	715	−0.29 (−0.50 to −0.07)	44
IL-10	11	629	0.52 (0.04 to 1.00)	87
TNF-α	11	558	−0.48 (−0.84 to −0.12)	74
**Aerobic interventions**				
CRP	10	383	−0.23 (−0.74 to 0.01)	65
IL-6	7	296	−0.20 (−0.48 to 0.08)	28
IL-10	5	239	0.28 (−0.31 to 0.87)	78
TNF-α	3	122	−0.34 (−0.84 to 0.16)	55
**Resistance interventions**				
CRP	7	269	−0.39 (−0.69 to −0.09)	27
IL-6	6	361	−0.39 (−0.78 to 0.01)	66
IL-10	6	390	0.69 (0.00 to 1.37)	89
TNF-α	5	359	−0.72 (−1.20 to −0.23)	77
**Combined interventions**				
CRP	6	280	−0.05 (−0.29 to 0.18)	0
IL-6	3	58	−0.16 (−0.68 to 0.36)	0
IL-10	0	0	-	-
TNF-α	2	37	0.16 (−0.49 to 0.81)	0

IL = interleukin; TNF-α = Tumor necrosis factor-alpha; CRP = C reactive protein; CI = confidence interval; SMD: standard mean difference.

## Data Availability

Data are available upon request due to privacy.

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
