# Peer review of "Effects of Exercise on Inflammatory Markers in Individuals with Chronic Kidney Disease: A Systematic Review and Meta-Analysis"

_metabolites, 2023, doi:10.3390/metabo13070795_

Round 1

Reviewer 1 Report

The presented article is an extensive study dealing with the evaluation of the effect of exercise on inflammatory markers in individuals with CKD. The manuscript is well written, the authors presented a lot of results. Although the authors summarized the data from many other researchers, this should not preclude its acceptation for publication. The evaluation of inflammatory markers and mediators, including the diagnostically most important acute phase proteins and cytokines during systemic inflammatory states is of great importance to detect uncontrolled inflammatory processes and their modulation. Therefore, the results of the presented manuscript may provide useful information regarding the assessment of the effect of exercise on inflammatory markers.

The main goal of this manuscript was to prepare a systematic review about the effect of exercise on inflammatory markers in patients with CKD, and these objectives were met.  

The topic of this study is relevant in the field, providing useful data also for practitioners with possible clinical applicability. The presented results showed that exercise interventions may be effective in the modulation of inflammatory processes, which could be important in the prevention of uncontrolled inflammatory reactions also in patients with CKD. These aspects of the clinical applicability of the obtained results should added into conclusions.

The methodology is adequate, the authors used electronic procedures for searching available papers dealing with the given issue and applied meta-analysis. The references are relevant and the number of cited references indicate that the authors have an overview in the given issue.  

I have no specific comments to the authors; and as I mentioned above, my only recommendation is to complete the conclusions with regard to the clinical applicability of the obtained results. 

Author Response

Dear reviewer,

Many thanks for the constructive comments. We had already written such clinical applicability in our conclusions.

Text: ". Thus, resistance exercise interventions must be recognized as the primary exercise type to be prescribed when targeting anti-inflammatory effects in individuals with CKD."

Reviewer 2 Report

In this review, the authors underlying the positive effects of exercise in individuals with chronic kidney disease (CKD), in particularly the authors, evaluate the effects of the exercise on inflammatory markers in the whole spectrum of CKD.

The manuscript is well articulated, with clear diagrams and tables, finally, the review is well written and reads well.

So I conclude for Accept in present form.

none

Author Response

Dear reviewer,

Many thanks for taking your valuable time to read our research paper.

Reviewer 3 Report

The paper entitled  “Effects of Exercise on Inflammatory Markers in Individuals 2 with Chronic Kidney Disease: A Systematic Review and Meta-3 Analysis” written by Baião and colleagues is a systematic review that analyzes the effect of the exercise program in patients with CKD. The paper is well done and describes interesting cytokine pattern modulation by exercise.

1.    According to the instructions for authors, the abstract should be a maximum of 200 words. The current abstract has 253 words. Please change it.

2.    Add a space between the references and the preceding text in line 229: …the IL-6 levels(10,17)…

3.    In this paragraph (lines 286-294) “Furthermore, our meta-analysis showed a significant reduction in IL-6 after pooled exercise interventions, which might be positive, because higher IL-6 values are associated with the progression of coronary artery calcification and mortality incident in dialysis individuals (59). IL-6 is considered a cytokine acutely anti-inflammatory and chronically pro-inflammatory and its levels are increased in many inflammatory diseases (60). Also, the muscle mechanical stress generated by exercise acutely increase the IL-6 levels(10,17) and this is followed by an increase in anti-inflammatory cytokines such as IL-1ra and IL-10 (17,61). Thus, we believe that the decrease of the IL-6 values may lead to a chronic improvement in the inflammatory condition in individuals with CKD”, the beneficial role of reduced IL-6 in CKD patients is not clear. Please clarify.

Author Response

Dear reviewer,

Thanks for carefully reading our research paper and making constructive suggestions.

  • We have adequately rewritten the Abstract to fit into the 200 words:

"Abstract: Individuals with chronic kidney disease (CKD) have a systemic inflammatory state. We assessed the effects of exercise on inflammatory markers in individuals with CKD. Electronic search was conducted, including MEDLINE. Experimental clinical trials that investigated the effects of exercise on inflammatory markers in individuals with CKD at all stages were included. Me-ta-analyses were conducted using the random effects model and standard mean difference (SMD). Subgroup analyses were performed for resistance, aerobic, and combined exercise interventions. Twenty-nine studies were included in the meta-analyses. Exercise interventions showed significant reductions in C-reactive protein (CRP) (SMD:-0.23; 95%CI:-0.39 to -0.06), interleukin (IL)-6 (SMD:-0.35; 95%CI -0.57, -0.14) and tumor necrosis factor-alpha (TNF-α) (SMD:-0.63, 95%CI:-1.01, -0.25) when compared with the controls. IL-10 levels significantly increased (SMD:0.66, 95%CI:0.09, 1.23) with exercise interventions. Resistance interventions significantly decreased CRP (SMD:-0.39, 95%CI:-0.69, -0.09) and TNF-α (SMD:-0.72, 95%CI: -1.20, -0.23) levels, while in-creasing IL-10 levels (SMD:0.57, 95%CI: 0.04, 1.09). Aerobic interventions only significantly reduced IL-6 levels (SMD:-0.26, 95%CI: -0.51, -0.01). No significant changes in any inflammatory markers were observed with combined exercise interventions. Exercise interventions are effective as an anti-inflammatory therapy in individuals with CKD compared to usual care control groups. Resistance interventions seem to promote greater anti-inflammatory effects."

  • We have added the space.

  • We have rewritten to make it clearer:

"Our meta-analysis showed a significant reduction in IL-6 after pooled exercise interventions. IL-6 is considered an acute anti-inflammatory cytokine, whereas chronically it may be pro-inflammatory (59). Muscle mechanical stress generated by acute bouts of exercise increases IL-6 levels (10,17), however; this is followed by an increase in anti-inflammatory cytokines such as IL-1ra and IL-10 (17,60). Previous evidence has shown that high IL-6 values are associated with the progression of coronary artery calcification and mortality in dialysis individuals (61). Thus, we believe that the decrease of IL-6 values promoted by exercise interventions may lead to a chronic improvement in the inflammatory condition in individuals with CKD (10,62) as well as a protective effect on cardiovascular outcomes, such as coronary artery calcification."